# Investigation of MicroRNA Biomarkers in Equine Distal Interphalangeal Joint Osteoarthritis

**DOI:** 10.3390/ijms232415526

**Published:** 2022-12-08

**Authors:** Melissa E. Baker, Seungmee Lee, Michael Clinton, Matthias Hackl, Catarina Castanheira, Mandy J. Peffers, Sarah E. Taylor

**Affiliations:** 1The Roslin Institute, The University of Edinburgh, Easter Bush, Roslin EH25 9RG, UK; 2The RICE Group, Division of Gene Function and Development, Roslin Institute, The University of Edinburgh, Easter Bush, Roslin EH25 9RG, UK; 3TAmiRNA GmbH, Leberstraße 20, 1110 Vienna, Austria; 4Department of Musculoskeletal and Ageing Science, Institute of Life Course and Medical Sciences, The University of Liverpool, William Henry Duncan Building, 6 West Derby Street, Liverpool L7 8TX, UK

**Keywords:** microRNA, biomarkers, synovial fluid, equine, distal interphalangeal joint, osteoarthritis

## Abstract

Osteoarthritis of the equine distal interphalangeal joint is a common cause of lameness. MicroRNAs from biofluids are promising biomarkers and therapeutic candidates. Synovial fluid samples from horses with mild and severe equine distal interphalangeal joint osteoarthritis were submitted for small RNA sequencing. The results demonstrated that miR-92a was downregulated in equine synovial fluid from horses with severe osteoarthritis and there was a significant increase in COMP, COL1A2, RUNX2 and SOX9 following miR-92a mimic treatment of equine chondrocytes in monolayer culture. This is the first equine study to evaluate the role of miR-92a in osteoarthritic chondrocytes in vitro.

## 1. Introduction

Musculoskeletal disorders are the most prevalent health problem in ageing horses [1]. Osteoarthritis (OA) is reported to be the most common cause with a prevalence of greater than 50% in horses over 15 years and greater than 80% in horses over 30 years [2]. OA is a disease of diarthrodial joints characterised by the progressive degradation of articular cartilage in addition to synovitis, subchondral bone sclerosis and osteophyte formation [3,4]. High-motion joints of the equine distal limb are significantly affected by OA [3,5,6] with the distal interphalangeal joint (DIPJ) being commonly affected [7]. OA is also a prevalent disease in humans with over 240 million sufferers worldwide [8] and estimated healthcare costs equivalent to 1–1.5% of the gross domestic product of developing countries [9]. There are no diagnostic tools to diagnose the early onset of the disease in either human or veterinary medicine. The best detection methods available are in the form of advanced imaging [10,11]; however, only later stages of the disease can be detected when substantial and irreversible cartilage degeneration has occurred.

Molecular biomarkers offer great potential for early disease diagnosis and there has been considerable research into microRNAs (miRNAs) over the past two decades. miRNAs are a class of small noncoding RNAs consisting of 19–24 nucleotides that regulate gene expression at the post-transcriptional level by primarily inhibiting translation or increasing target messenger RNA (mRNA) degradation by binding to the 3′ untranslated regions (UTR) of target transcripts [12]. However, they can also directly interact with the promoter and activate gene expression through RNA activation [13]. Each miRNA can regulate hundreds of genes in any given cell type and they participate in almost every aspect of cellular physiology; differentiation, metabolism, proliferation and apoptosis [12]. Therefore, during cellular or tissue damage miRNA expression profiles change and can be associated with specific diseases [14,15,16].

The stability of miRNAs in biofluids, such as serum, plasma, saliva, urine, milk and synovial fluid [17,18,19,20,21], make miRNAs promising diagnostic candidates, with the added advantage that most biofluids can be obtained by non-invasive methods. A study investigating potential biomarkers for OA and rheumatoid arthritis in human synovial fluid and plasma samples found the expression patterns of miRNAs in OA synovial fluid were similar to the expression in synovial tissues and were significantly lower than their plasma concentrations [22]. The findings suggested that the miRNAs identified were likely to originate from synovial tissues. Similarly, another study examining serum and synovial fluid from human OA patients demonstrated that serum-derived miRNAs did not fully represent the miRNAs identified in synovial fluid [23]. Biofluid miRNA research in equine OA is extremely limited. In 2019, a pilot study reported a robust repeatable methodology for miRNA isolation from equine synovial fluid and plasma [24]. The first equine study to investigate small non-coding RNA signatures in early OA of equine synovial fluid found miR-223 was significantly downregulated in synovial fluid from the early OA group compared to the controls and miR-23b was significantly upregulated [25]. Furthermore, another equine study investigating small non-coding RNA extracellular vesicles in a post-traumatic OA model of the middle carpal joint discovered miR-23a was upregulated in the OA group compared to the control group [26]. Several human studies have demonstrated miR-23b is upregulated in OA cartilage compared to normal cartilage samples, with authors concluding the downregulation of miR-23b could be a potential therapeutic strategy for the treatment of OA [27,28,29].

The differential expression of miRNAs in synovial fluid offers excellent biomarker potential for early OA diagnosis and may lend itself to treatment development. We hypothesized that the miRNAs would demonstrate different differential expression profiles in mild OA synovial fluid samples compared to severe OA samples of the equine DIPJ.

## 2. Results and Discussion

### 2.1. Samples

Synovial fluid from seventeen horses underwent small RNA sequencing and were later used for RT-qPCR (real-time quantitative reverse transcription polymerase chain reaction) validation. Table 1 details the sex, mean weight, mean age, breed and reason for euthanasia of the donor horses used for synovial fluid sample collection.

Cartilage was collected from the DIPJ of three donor horses in the synovial fluid biomarker small RNA sequencing study to grow chondrocytes in a monolayer for future experiments; H3, H16 and H17. Cartilage was also collected from three donors not included in the biomarker study; H26, H27 and H28. Table 2 details the breed, age, sex, weight, reason for euthanasia and OA group of each donor horse.

### 2.2. Small RNA Sequencing Analysis

Small RNA sequencing detected between 195–266 miRNAs per sample. In total, 382 different miRNAs were detected in the synovial fluid samples across all samples (excluding spike-ins) (Appendix A). The RPM (reads per million) after normalisation ranged from 0.7–324,034.45 RPM with most miRNAs having <50,000 RPM. Unsupervised clustering and visualisation as a heatmap was performed to help visually identify patterns and relationships between the two groups as well as outliers. However, the analysis revealed no outlying samples and largely overlapping miRNA profiles between the mild and severe OA groups (Figure 1). Low-abundant miRNAs were filtered from the dataset prior to differential expression analysis using the independent filtering procedure implemented in DeSeq2 [30]. Three miRNAs; miR-25, miR-16 and miR-92a, were found to be significantly (FDR < 0.1) differentially expressed (DE) and downregulated in the severe OA synovial fluid samples compared to the mild synovial fluid samples, appreciated in the volcano plot in Figure 2, which shows FDR (false discovery rate) and logFC (log fold change) for this comparison. The logFC, *p* value, FDR and mean RPM for each DE miRNA in the mild and severe OA group are shown in Appendix A and the logarithmic plots of RPM for each miRNA calculated using edgeR’s glmQLFTest method are shown in Figure 3.

### 2.3. Validation of Small RNA Sequencing by RT-qPCR

The RPM of all miRNAs detected in the synovial fluid samples were inputted into the software programme Normfinder and miR-27b was found to have the most consistent read count throughout the samples (mean RPM 10,000) with the best stability value and standard error (Appendix A). Therefore, miR-27b was used as the normaliser for RT-qPCR analysis of synovial fluid. Based on the logFC, *p* value, FDR and mean RPM for each DE miRNA and power calculations (Appendix A) miR-92a and miR-16 were chosen for validation. miR-25 was not included for validation as all the samples in the mild OA group were within the same RPM range as the severe OA group samples apart from one major outlier at 160,000 RPM (Figure 3C) which skewed the results. Although miR-92a and miR-16 were shown to be significantly different between groups in the small RNA sequencing data, RT-qPCR did not validate sequencing findings and there was no difference between groups. The results of miR-92a and miR-16 validation are shown in Figure 4. The outlier in the mild OA group for both miR-92a and miR-16 was the same sample, synovial fluid from H8 which also showed the highest RPM count for both miRNAs in the small RNA sequencing analysis. Retrospective evaluation of this sample highlighted no significant findings in the clinical history, no differences in sample handling and preparation and no major differences in macroscopic and microscopic scores to the other horses in the mild OA group.

### 2.4. Gain and Loss Function Studies for miR-92a

Although a significant difference in miR-92a expression between the mild and severe OA synovial fluid samples was not demonstrated by RT-qPCR, miR-92a was DE expressed in equine synovial fluid and it has been identified as an important miRNA in the pathogenesis of OA in other species; therefore, it was decided to investigate this miRNA further. Ingenuity Pathway Analysis (IPA) (IPA, Qiagen Redwood City, CA, USA) was used to identify miR-92a mRNA target genes. We used a conservative filter, experimentally validated and highly conserved predicted mRNA targets for miR-92a. The miRNA Target Filter in IPA provides insights into the biological effects of miRNAs, using experimentally validated interactions from TarBase and miRecord. Selected targets were measured in the five treatments; un-transfected chondrocytes, lipofectamine only, miR-92a mimic, miR-92a inhibitor and negative control. Un-transfected and lipofectamine chondrocytes were additional experimental control measures to ensure the lipofectamine system did not have toxic effects on the equine chondrocytes. The results of the target mRNA genes measured are shown in Figure 5: COMP (A), COL1A2 (B), RUNX2 (C), SOX9 (D), ACAN (E) and COL3A1 (F). There was a significant (*p* < 0.05) increase in COMP, COL1A2, RUNX2 and SOX9 expression in the miR-92a mimic treatment compared to the negative control. The expression of COMP, COL1A2 and RUNX2 in the inhibitor treatment was similar to the negative control however expression of SOX9 was higher in the inhibitor treatment compared to the negative control, although this was not statistically significant. There was little difference in ACAN and COL3A1 expression between the miR-92 mimic and miR-92a inhibitor treatment in all horses. COL2A1, MMP-13, ADAMTS-5 and ADAMTS-4 were also measured in all five treatments but showed little to no expression.

#### Discussion

In this study, we used a non-biased global approach to screen equine synovial fluid samples from OA DIP joints for miRNA expression which has not been investigated previously. The research used two cohorts of horses, a mild OA and a severe OA cohort, which were grouped by macroscopic scoring and gold standard microscopic histological scoring of cartilage samples taken from the joint. No lameness examination was undertaken, it may have been an interesting comparison to determine the level of lameness for each horse; however, the level of lameness is not directly linked to the level of degenerative changes in a joint [31,32]. A control population would have been ideal; however, none of the horses in our study had completely macroscopically and microscopically normal DIPJs, demonstrating the difficulty of obtaining a control population in clinical research. The sex, weight, breeds and reason for euthanasia were very similar in both groups with the main difference being the mean age, 9 ± 3 years in the mild OA group and 18 ± 9 years in the severe OA group, which is to be expected as OA is related to ageing [33,34]. The objective of the research was to identify DE miRNAs in mild and severe OA DIPJ synovial fluid. The small RNA sequencing detected 384 miRNAs in total (Appendix A) which was higher than previously reported in the equine synovial fluid where a total of 323 small non-coding RNAs, miRNAs, snoRNAs and snRNAs, were identified [25]. This may represent the joint differences between the DIPJ and metacarpophalangeal joint.

The results found that three miRNAs (miR-92a, miR-16 and miR-25) were significantly (FDR < 0.1) upregulated in the synovial fluid collected from horses with mild DIPJ OA compared to severe DIPJ OA. A post-traumatic equine carpal OA experimental model study also identified miR-92a in equine synovial fluid and reported miR-92a expression increased in extracellular vesicles derived from synovial fluid in early OA along with a panel of five other miRNAs (miR-23a, miR-25, miR-215, miR-486-5p and miR-451) [26]. Another equine study profiling miRNA in synovial fluid from the metacarpophalangeal joint showed miR-223 was significantly downregulated and miR-23b was significantly upregulated in the early OA group compared to the control group [25]. Both miR-223 and miR-23b were detected in our small RNA sequencing but showed no significant differences between the two groups. This may reflect population differences; control and early OA compared to mild and severe OA. There is a wealth of literature on miRNA synovial fluid profiles in human OA which demonstrate a wide range of profiles [22,23,35,36,37]. Li et al. 2016 collected synovial fluid from early and late knee OA patients and identified a miRNA profile (miR-335-3p, miR-199a-5p, miR-671-3p, miR-1260b, miR-191-3p, miR-335-5p and miR-543) that distinguished the two groups [38]. The profile reported differs from the results of the current study which is probably attributed to numerous factors including species differences, individual joint-related changes [39] and distinctive severities of OA. The OA disease stage is likely to influence the cell type excreting free and extravesicular miRNAs, e.g., chondrocytes, synoviocytes, osteoclasts or osteocytes depending on which tissue is most affected during the OA phase, with several pathways being proposed to cause the release of miRNAs from cells [40,41]. Furthermore, Li et al. 2016 collected synovial fluid from symptomatic patients with radiographic evidence of OA whereas the equine synovial fluid in the current study was selected based on gross pathology and histopathological changes.

RT-qPCR was used as platform to validate the results for miRNA with a good read count of >10,000 RPM and an FDR value of <0.1. RT-qPCR validation of increased expression of miR-92a in mild OA was not in agreement with the findings of the small RNA sequencing. This is most likely because RT-qPCR is not as sensitive as small RNA sequencing and the fold-change difference between the RPM in each group was not large enough to demonstrate a difference by RT-qPCR. Everaert et al., 2017, compared five RNA sequencing analysis pipelines to wet-lab qPCR results for >18,000 protein-coding genes and showed 15–20% of genes showed differential expression in opposing directions, or one method showed DE whilst the other did not [42]. Furthermore, a power calculation using our small RNA sequencing data for miR-92a showed a sample size of *n =* 36 would be needed to demonstrate a power of 95% (significance level = 0.05) between the two groups (Appendix A). In our current study, we used five mild OA and twelve severe OA synovial fluid samples. The sample number for the mild OA cohort was smaller than the severe OA cohort which is an unfortunate limitation of this clinical research. The study involved a collection of samples from a mixed population of horses donated to an equine hospital in a set period and there was no control over which OA group their DIPJs would be classified to. The power calculation has demonstrated that a larger sample size of at least thirty-six horses in each group should be used for future research.

We identified that miR-92a was decreased in synovial fluid from horses with more severe OA. We, therefore, wanted to investigate the role of miR-92a in OA chondrocytes, as cartilage is one of the main tissues degraded during OA and miR-92a was expressed at good levels (low Ct values between 20–26) when measured in the equine cartilage tissue and chondrocytes. As miRNAs affect gene expression of their target genes, we used IPA (IPA, Qiagen Redwood City, CA, USA) to identify miR-92a mRNA target genes. These included ACAN, COL2A1, COMP, SOX9, RUNX2, ADAMTS-4, ADAMTS-5 and MMP-13, all important genes involved in the pathogenesis of OA [43,44]. The relationships between these targets and miR-92a have been investigated previously and experimentally validated in human and mouse chondrocytes in vitro [45,46,47] and in cartilage tissue in an in vivo mouse study [47]. Therefore, together with our small RNA sequencing data identifying miR-92a as a potential biomarker of OA severity in synovial fluid, we chose to undertake gain and loss of function experiments in equine chondrocytes. We demonstrated potential roles for miR-92a in equine OA through these experiments with increased expression in chondrogenic markers; COMP and SOX9 as well as hypertrophic markers COL1A2 and RUNX2 in chondrocytes transfected with the miR-92a mimic. In human OA cartilage, miR-92a was decreased compared to normal cartilage [47] and, interestingly, the miR-92a levels in the equine synovial fluid from severe DIPJs were decreased compared to the mild OA samples. If control synovial fluid could have been obtained, then they would have most likely had the highest levels of miR-92a. The same study also reported overexpression of miR-92a in human OA chondrocytes resulted in increased mRNA expression of ACAN, COMP, COL2A1 and SOX9. Furthermore, the study’s in vivo tibiofemoral collagenase-induced OA mouse model demonstrated an increase in COL2A1 and ACAN mRNA and protein expression in the miR-92a mimic treated group compared to the controls and histological analysis showed that the severity of femoral articular cartilage deterioration was also the mildest in this group. The same authors demonstrated overexpression of miR-92a resulted in decreased expression of the important aggrecanases involved in cartilage degradation, ADAMTS-4 and ADAMTS-5, in primary human chondrocytes that were stimulated with IL-1β [45]. There was little to no expression of ADAMTS-4 and ADAMTS-5 in the equine chondrocytes cultured (Ct values > 36 or no Ct value recorded). Our current study is in agreement with increased expression of COMP and SOX9 in human knee OA chondrocytes [47]; conversely, our results showed an increase in RUNX2 with the miR-92a mimic treatment compared to a decrease seen in murine chondrocytes and failed to detect measurable levels of COL2A1 (Ct values > 36 or no Ct value recorded). ACAN expression was detectable (Ct values ranged from 26–28); however, the expression was similar in all treatments. The differences in COL2A1 expression are likely due to equine chondrocyte dedifferentiation in monolayer [48]. This has been reported previously and research on equine chondrocytes obtained from healthy articular cartilage of metacarpophalangeal joints investigated how in vitro culture influences differentiation of equine articular chondrocytes by analysing chondrogenic markers from passage 0 (P0) to passage 8 (P8) [48]. The chondrocytes underwent a substantial morphological and phenotypic change with dedifferentiation starting early in culture (P0-P1) shown by a decrease in COL2A1 expression in all groups, with both markers reaching very low levels at P3. COL1A2 increased from P0, reaching the highest expression at P3. COL1A2 has been shown to be a major marker of chondrocyte dedifferentiation and is a well-characterised fibroblastic marker [49,50,51]. The elderly horses showed dedifferentiation shortly after isolation with reduced proliferative capacity and the lowest COL2A1 expression at P0 compared to the other groups. Three out of six of the donor horses in the current study were harvested from elderly horses (22–28 years old) and all cells were isolated from cartilage tissue with OA. Results from the six donors showed high COL1A2 expression and no COL2A1 expression indicating dedifferentiation into a fibroblastic phenotype [49,50,51]. This, therefore, will have influenced the results of the miR-92a transfection experiments as the cells had a fibroblastic phenotype rather than a chondrogenic phenotype. Furthermore, RUNX2 expression was increased after the miR-92a mimic treatment compared to a decrease which has been demonstrated in murine chondrocytes [47]. This could also be attributed to the equine cell phenotype in monolayer, as RUNX2 is a hypertrophic marker and the murine chondrocytes investigated retained a chondrogenic phenotype. Experiments were designed to use chondrocytes from donors with OA to investigate the role that miR-92a could play in diseased equine chondrocytes. However, a different in vitro approach is needed to ensure the cells retain their chondrogenic phenotype so miR-92a can be reliably investigated. The donor cells were expanded in monolayer, passaged and collected at P1 to be stored at −80 °C. The cells were then thawed to be grown in T75 flasks and passage into 6-well plates before transfection. The additional two passaging steps will have contributed to dedifferentiation and unfortunately this was a limitation of clinical sample collection. COL1A2 expression was high in all five treatment measures but interestingly expression significantly increased in the miR-92a mimic treatment which has not been reported in human and murine in vitro studies and demonstrates a relationship pathway between miR-92a and COL1A2 in equine chondrocytes.

Collectively all six donors, each with three biological replicates showed a significant increase in COMP, COL1A2 and SOX9 expression in the miR-92a mimic treatment compared to the negative control. The results suggest that miR-92a in equine chondrocytes is likely acting on the same HDAC2 pathway that has been established in primary human OA chondrocytes, where overexpression of miR-92a caused significant downregulation of HDAC2 and increased cartilage-specific gene expression such as COMP and SOX9 through interaction with the 3′-UTR of HDAC2 mRNA determined through a luciferase reporter assay [46]. The opposite effect was seen with miR-92a inhibitor treatment which significantly upregulated HDAC2, decreasing cartilage-specific gene expression. In our study, the expression of the target mRNA genes in the miR-92a inhibitor treatment was very similar to the negative control treatment. The miRNA inhibitors are designed to specifically bind to and inhibit endogenous miRNA to inhibit their function. The results suggest that inhibiting endogenous miR-92a does not significantly downregulate the target mRNA measured in the study, expression levels remain similar to that of the negative control chondrocytes. This is in contrast to what has been demonstrated in human and murine chondrocytes [46,47] which may be because other miR-92 family members are involved in the target gene regulation such as miR-92b, which is a known miRNA in the horse genome [52]. It is possible that miR-92b alongside miR-92a also interacts with the 3′-UTR of HDAC2 mRNA in the equine chondrocyte and so additional transfection experiments could be performed with both miR-92a and miR-92b inhibitors to see whether this causes significant downregulation of the target mRNA.

The standard deviation range was fairly large between donors and this may have been due to randomly sampling cells with different molecular profiles. Other work using a single-cell RNA sequencing approach has defined nine distinct molecular profiles in healthy murine cartilage that altered with early disease [53]. Therefore, it is possible that the donors had slightly differing chondrocyte subtypes which influenced gene expression to different degrees. Furthermore, the cartilage was sampled from donors which all had OA but were not necessarily in the same stages of the disease. The results did not show that any particular breed influenced the results of the in vitro chondrocyte experiments or miRNA expression in the small RNA sequencing data. This is supported by findings from other equine miRNA studies using mixed breed populations which have shown no correlation between breed and miRNA expression [25,54]. However, further research using a larger mixed-breed sample size would be beneficial and strengthen the data reported in this initial study.

Future work should proceed to proteomic studies to establish the protein levels of the mRNA genes investigated in this study in equine chondrocytes after miR-92a transfection and develop an in vitro equine model that permits chondrocytes from diseased tissue to maintain their chondrogenic phenotype whilst allowing miRNA transfection. Studies in human [55] and porcine [56] chondrocytes using 3D pellet and alginate-based systems have been successful in retaining a chondrogenic phenotype demonstrated by high expression of chondrogenic markers however, these systems have limitations as monolayer culture is required for cell transfection. Concentrated research in this area will also aid cell-based cartilage repair strategies, which are currently suboptimal in the horse [57] and potentially allow in vivo application in the future.

## 3. Methods and Materials

### 3.1. Sample Collection

Samples were collected from clinical cases at the R(D)SVS, Equine Hospital, University of Edinburgh which were euthanised for clinical disease between 2019–2022. All cases included in the study had owners’ consent for research purposes. Ethical approval was provided by the veterinary ethical research committee (VERC) of the University of Edinburgh (VERC approval ref: 9314 and 4.18).

The skin was aseptically prepared and a 20 G 1.5-inch needle was inserted into the DIPJ using a dorsal or dorsolateral approach, synovial fluid was aspirated and transferred into Eppendorf tubes within one hour of euthanasia. The Eppendorf tubes were centrifuged at 4 °C for 10 min at 3000× *g* to remove cellular debris, the supernatant was extracted and stored at −80 °C.

### 3.2. Macroscopic and Microscopic Cartilage Assessment

Each DIPJ was opened by performing a circumferential cut above the coronary band and dissection through the collateral ligaments using a No.10 scalpel blade. The articular surfaces of the second (P2) and third (P3) phalanges were inspected and graded using an adapted DIPJ scoring system from McIlwraith et al. macroscopic staging technique for the metacarpo/metatarsophalangeal joint [58]. There are no reports of a specific articular cartilage macroscopic grading system for the equine DIPJ in the literature; therefore, a system was developed and validated to grade the joints in the study. Macroscopically normal and abnormal cartilage and subchondral bone were sampled using a cast saw. Specimens were fixed in 10% neutral buffered formalin and then immersed in Histo-Decal^®^ for 7–10 days and embedded in paraffin wax. Four-micron sections were cut and stained with haematoxylin and eosin and safranin O to assess the cartilage and proteoglycan content. Sections were examined and graded using the OARSI (Osteoarthritis Research Society International) scoring system [59]. Based on the macroscopic and microscopic assessments donors were assigned to a mild OA group (*n =* 5) or a severe OA group (*n =* 12). The horse was classified as mild OA if the total gross joint grade of both the two feet per horse was <−6 and the combined OARSI grade of both feet was ≤3. The horse was classified as severe OA if the total gross joint grade of both the two feet per horse was ≥7 and the combined OARSI grade of both feet was ≥3. Figure 6 shows an example of the DIPJ macroscopic appearance and the corresponding histological sample of one horse in the mild and one horse in the severe group.

### 3.3. Total RNA Extraction, Library Preparation and Small RNA Sequencing

Total RNA from the synovial fluid samples was extracted using the miRNeasy Serum/Plasma kit (Qiagen, Crawley, UK). Due to low RNA concentrations commonly observed in biofluids, a fixed volume of 8.5 µL was used for library preparation using RealSeq Biofluids Plasma/Serum miRNA Library Kit (RealSeq Biosciences, Santa Cruz, CA, USA). The PCR cycles for library amplification were optimized during a pilot study and selected as 22 cycles. Library size and concentration underwent quality control using the Bioanalyzer DNA1000 chips (Agilent, Santa Clara, CA, USA). Libraries were diluted and pooled at an equimolar rate before size purification using a 3% Agarose Cassettes for the BluePippin (Sage Science, Beverly, MA, USA) and a size range of 130–160 bp. The resulting pooled library was sequenced on an Illumina NextSeq 550 platform in high-output mode with 75 bp single-end reads.

### 3.4. Small RNA Sequencing Data Analysis

Data were analysed using the miND pipeline. The overall quality of the next-generation sequencing data was evaluated automatically and manually with fastQC v0.11.8 [60] and multiQC v1.7 [61]. Reads from all passing samples were adapter trimmed and quality filtered using cutadapt v2.3 [62] and filtered for a minimum length of 17 nt. Mapping steps were performed with bowtie v1.2.2 [63] and miRDeep2 v2.0.1.2 [64], whereas reads were mapped first against the genomic reference EquCab.3.0 provided by Ensembl [65] allowing for two mismatches and subsequently miRBase v22.1 [66], filtered for miRNAs of eca only, allowing for one mismatch. For a general RNA composition overview, non-miRNA mapped reads were mapped against RNAcentral [67] and then assigned to various RNA species of interest. Statistical analysis of pre-processed NGS data was done with R v3.6 and the packages heatmap v1.0.12, pcaMethods v1.78 and genefilter v1.68. Differential expression analysis with edgeR v3.28 [68] used the quasi-likelihood negative binomial generalized log-linear model functions provided by the package. The independent filtering method of DESeq2 [69] was adapted for use with edgeR to remove low abundant miRNAs and thus optimize the FDR correction. Significantly differentially expressed miRNAs were determined as *p* < 0.05.

### 3.5. Small RNA Sequencing Validation

RT-qPCR was used to validate two significant DE miRNA detected in the small RNA sequencing. The seventeen synovial fluid samples were treated with 1 µg/ul working hyaluronidase solution to reduce the viscosity and incubated at 37 °C for 60 min then centrifuged at 1000× *g* for 5 min. The supernatant was transferred to a Corning^®^ Costar^®^ Spin-X^®^ tube filter, pore size 0.22 µm and centrifuged at 5000× *g* for 15 min. RNA was extracted from 200 µL using the miRNeasy Serum/Plasma kit (Qiagen, Crawley, UK). The reverse transcription was performed using the miRCURY LNA RT kit (Qiagen, Crawley, UK). Samples were placed into Biometra Advanced 60 PCR Thermal Cycler (Analytik Jena GmbH, Jena, Germany) and incubated at 42 °C for 60 min followed by 95 °C for 5 min and immediately cooled to 4 °C. The miRCURY LNA SYBR^®^ Green PCR Kit and miRCURY LNA primers assays (Appendix A) (Qiagen, Crawley, UK) were used for RT-qPCR in an Mx3000 RT-PCR System (Agilent, Santa Clara, CA, USA). Small RNA sequencing data was inputted into Normfinder [70] to identify the most stable endogenous reference gene and relative expression levels were normalised to the geometric mean of miR-27b and calculated using the 2^−ΔΔCt method [71].

### 3.6. Primary Chondrocyte Collection, Isolation and Cell Culture

Cartilage was sampled from the DIPJ of three donor horses (H3, H16 and H17) in the study and three donors not included in the biomarker study (H26, H27 and H28). Cartilage from the articular surface was collected and diced into small sections of 1–2 mm^3^ in a Petri dish. The cartilage pieces were digested with 0.1% collagenase II (Gibco Life Technologies, Grand Island, New York, NY, USA) at 37 °C for 12 h. Cells were then cultured in T75 flasks until confluent and cryopreserved. The cell vials were removed from −80 °C and seeded into T75 flasks at a density of 3.75 × 10^5^ cells per cm^2^. Once the cells had reached 80% confluency in the T75 flasks the cells were passaged and transferred into 6-well plates at a density of 5 × 10^3^ cells per cm^2^ and transfection was undertaken once 60% confluent.

### 3.7. miRNA Cell Transfection

Chondrocytes from each donor were grown in biological triplicate in 6-well plates for each treatment measure. The chondrocytes were transfected with either a miR-92a mimic, miR-92a inhibitor or miRNA negative control (nonspecific miRNA) (Thermo Fischer Scientific, Waltham, MA, USA) at a concentration of 50 nM. Lipofectamine^®^ RNAiMAX Transfection Reagent (Thermo Fischer Scientific, Waltham, MA, USA) was used to transfect cells according to the manufacturer’s instructions. Cells were harvested in QIAzol after 48 h for RT-qPCR.

### 3.8. RNA Extraction, Reverse Transcription and RT-qPCR of Equine Chondrocytes

RNA extraction was performed using the miRNeasy Mini Kit (Qiagen, Crawley, UK) with on-column DNase digestion following the manufacturer’s protocol after chondrocytes in each well were lysed with 700 µL QIAzol (Qiagen, Crawley, UK). RNA concentration and quality were determined using the NanoDrop ND1000 (Thermo Fischer Scientific, Waltham, MA, USA). The miRCURY LNA RT kit (Qiagen, Crawley, UK) was used for the reverse transcription of miRNA and the QuantiTect^®^ Reverse Transcription kit (Qiagen, Crawley, UK) for mRNA. The miRCURY LNA SYBR^®^ Green PCR Kit (Qiagen, Crawley, UK) was used for miRNA RT-qPCR and Platinum^®^ SYBR^®^ Green qPCR SuperMix-UDG (Thermo Fischer Scientific, Waltham, MA, USA) was used for mRNA RT-qPCR. Reactions were performed in triplicate. Transcript levels were normalized to the housekeeping gene glyceraldehyde 3-phosphate dehydrogenase (GAPDH) for mRNA or U6 RNA for miRNA. Primers used for the analyses are shown in Appendix A. Gene expression was calculated using the 2^−ΔΔCt method [71].

### 3.9. Statistical Analysis

Statistical evaluation of gene expression data, following normality testing, were performed using GraphPad Prism version 9.0 for Windows (GraphPad Software, La Jolla California USA, www.graphpad.com, accessed on 12 June 2022); *p* values are indicated. R version 4.1.2 (R Foundation for Statistical Computing, Vienna, Austria, www.R-project.org, accessed on 6 May 2022) with the ‘pwr’ package was used to calculate sample size power.

## 4. Conclusions

Small RNA sequencing demonstrated decreased expression of miR-92a in synovial fluid samples from horses with more severe OA in the DIPJ. Our gain and loss of function experiments demonstrated increased COMP, SOX9, RUNX2 and COL1A2 expression in the miR-92a mimic treatment compared to the negative control, suggesting miR-92a influences chondrogenic and hypertrophic markers important in OA in equine chondrocytes. Future research should focus on developing an equine in vitro chondrogenic model which supports dedifferentiation to help further define the role miR-92a plays in the equine chondrocyte during OA.

## Figures and Tables

**Figure 1 ijms-23-15526-f001:**
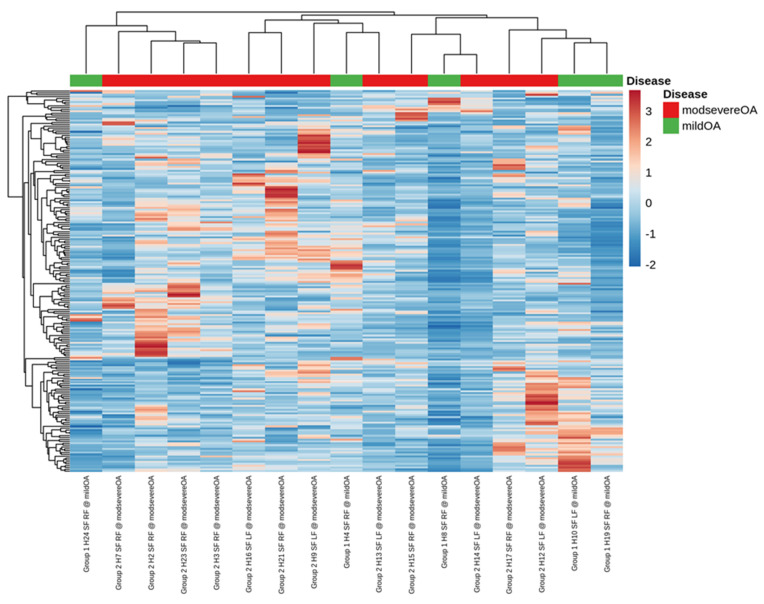
A visual representation of the data using a heatmap to identify patterns and relationships between the two groups as well as outliers. The analysis showed no outlying samples and largely overlapping miRNA profiles between the mild and severe osteoarthritis groups.

**Figure 2 ijms-23-15526-f002:**
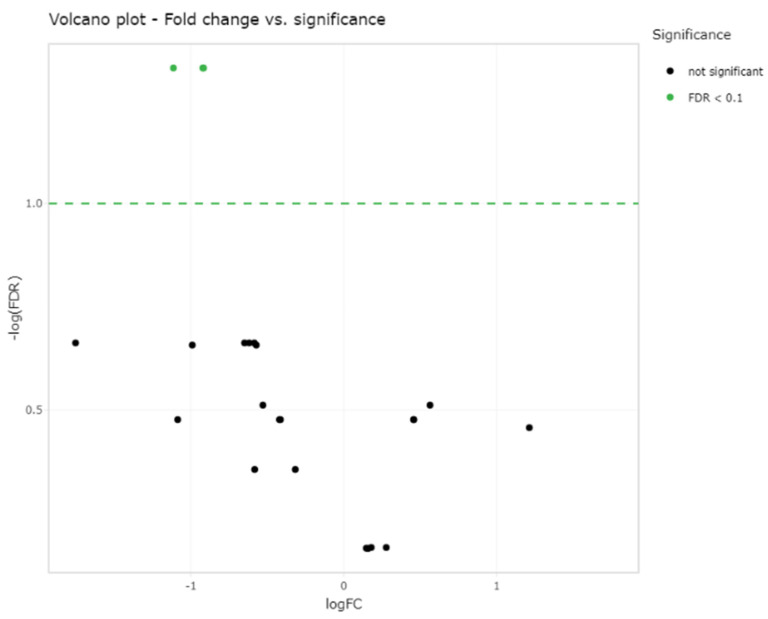
Volcano plot showing the relation of each miRNA logFC in the two groups and the statistical significance of this change (FDR < 0.1) which demonstrates three significantly differentially expressed miRNAs; miR-25, miR-16 and miR-92a (green dots) between the two groups (two green dots overlap (miR-16 and miR-92a)). *Log fold change (LogFC) and false discovery rate (FDR)*.

**Figure 3 ijms-23-15526-f003:**
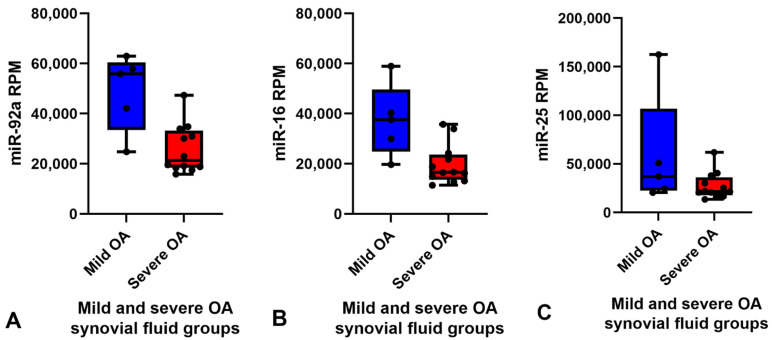
Box and whisker plots showing the RPM of each miRNA for each sample in the mild (blue) (*n =* 5) and severe (red) (*n =* 12) OA groups. Boxes show minimum RPM, median with interquartile range and maximum RPM. Significantly downregulated miRNAs in the severe OA synovial fluid group compared to mild OA group ordered by logFC (FDR < 0.1) (**A**) miR-92a, (**B**) miR-16 and (**C**) miR-25 calculated using edgeR’s glmQLFTest method. *Reads per million (RPM), log fold change (LogFC), false discovery rate (FDR) and osteoarthritis (OA)*.

**Figure 4 ijms-23-15526-f004:**
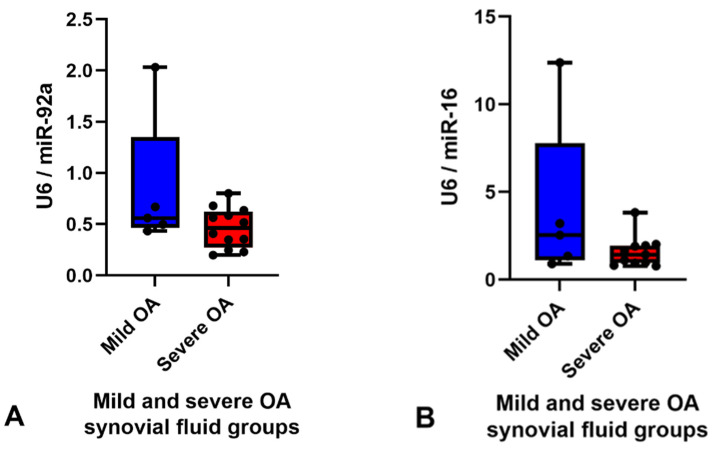
Graphs demonstrating expression levels of miR-92a (**A**) and miR-16 (**B**) in synovial fluid samples in the mild (blue) (*n =* 5) and severe (red) (*n =* 12) OA groups determined by RT-qPCR. The relative expression was calculated using 2^−ΔΔCt^ using miR-27b as the endogenous control. Boxes show minimum, median with interquartile range and maximum relative expression for each miRNA. Validation did not demonstrate any significant differences between groups, *p* < 0.05. *Osteoarthritis (OA)*.

**Figure 5 ijms-23-15526-f005:**
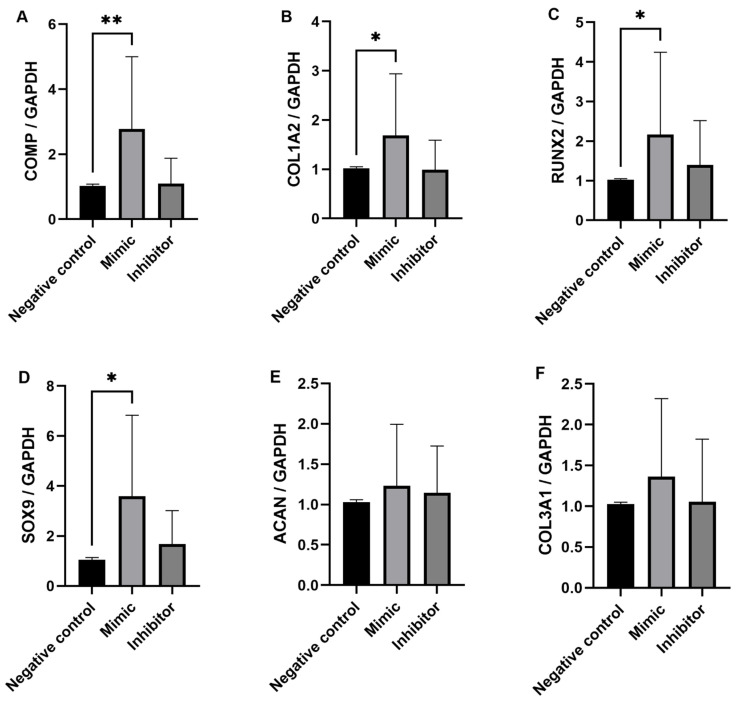
Expression levels of COMP (**A**), COL1A2 (**B**), RUNX2 (**C**), SOX9 (**D**), ACAN (**E**) and COL3A1 (**F**) determined by RT-qPCR with GAPDH as the endogenous control. Data is represented as means ± standard deviations of the three combined biological replicate samples for each treatment measure from each donor (*n =* 6). Statistical significance was tested in Graphpad Prism using a Mann–Whitney test. * *p* < 0.05, ** *p* < 0.005.

**Figure 6 ijms-23-15526-f006:**
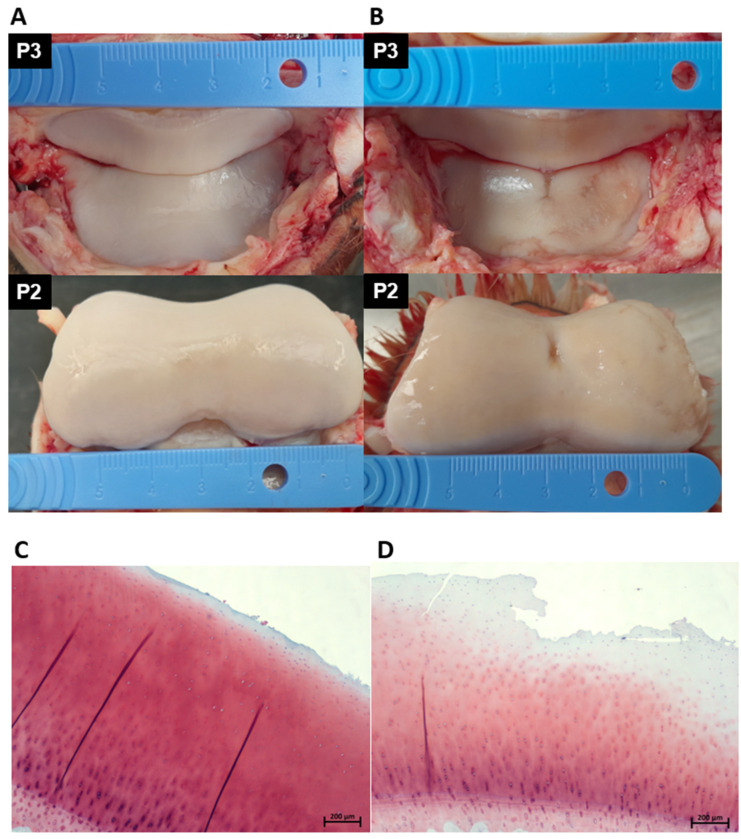
Photographs demonstrating the gross pathology of the articulating cartilage surface of P2 and P3 of a horse in the mild OA group (**A**) and a horse in the severe OA group (**B**). The horse in the mild OA group received a macroscopic score of 2/6 (**A**) and the horse in the severe group a macroscopic score of 5/6 (**B**). A corresponding histological section taken from the DIPJ is shown below the gross photographs; (**C**) safranin O-stained section showing mild surface fibrillation and microscopic cracks in the superficial zone; OARSI grade 1. (**D**) Safranin O-stained section showing severe surface disruption, loss of superficial layer matrix and loss of stain uptake in the superficial and middle layers indicating proteoglycan loss; OARSI grade 3. *DIPJ (distal interphalangeal joint), OARSI (Osteoarthritis Research Society International), third phalangeal bone (P3), second phalangeal bone (P2) and osteoarthritis (OA)*.

**Table 1 ijms-23-15526-t001:** Details of the seventeen donor horses used for synovial fluid sample collection.

	Mild OA Group	Severe OA Group
Sex	Three geldings Two mares	Six geldings Six mares
Mean weight	550 ± 70 kg	440 ± 70 kg
Mean age	9 ± 3 years	18 ± 9 years
Breeds	Highland Pony, Dutch Warmblood, Belgian Warmblood and Thoroughbred	Irish Draught, Welsh Pony, Icelandic Pony, Irish Sports Horse, Cob, Cross-breed pony, Thoroughbred, Sports Horse and Connemara Pony
Reasons for euthanasia	Colic, lameness, cervical vertebral malformation and one horse had a complete open third metacarpal bone fracture	Colic, lameness, incisional hernia, congestive heart failure, severe behavioural problems and advanced dental disease

**Table 2 ijms-23-15526-t002:** Details of the donor horses used for cartilage collection and monolayer cell culture.

Horse Number	Breed	Age (Years)	Sex	Weight (kg)	Reason for Euthanasia	Group
H3	Irish Draught	28	G	500 kg	Lameness	Severe OA
H16	Cross-breed horse	22	G	400 kg	Age-related concerns	Severe OA
H17	Cob-cross	26	G	400 kg	Age-related concerns	Severe OA
H26	Irish Sports Horse	5	M	500 kg	Multi-limb lameness	Mild OA
H27	Cob	7	M	500 kg	Grass sickness	Mild OA
H28	Irish Draught	12	G	600 kg	Lameness	Severe OA

## Data Availability

Data has been submitted to National Centre for Biotechnology Information; accession GSE205409. The datasets supporting the conclusions of this article are included within the article and its additional files.

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
