# Peer review of "Investigation of MicroRNA Biomarkers in Equine Distal Interphalangeal Joint Osteoarthritis"

_ijms, 2022, doi:10.3390/ijms232415526_

Round 1
Reviewer 1 Report
It is an exciting study and can be improved using more experimental datas.
Author Response
Thank you for taking the time to review our research article. Unfortunately, no further experimental data can be produced from this current study as the research involved collection of clinical samples over a set time period. Horses were donated by owners to an equine hospital over a 3-year period to allow collection of synovial fluid and cartilage tissue for cell culture experiments. We agree that future research involving a larger sample size of horses would be beneficial and strengthen the data produced in this initial study. A comment about the sample size has been added to the discussion, lines 278-284. Also section 2.1 of the Results and Discussion has been amended to make the results clearer for the reader and two tables have been added (lines 80-111) and more detail about the macroscopic scoring system has been added to the experimental methods section in 3.2.
Reviewer 2 Report
This manuscript used the small RNA-sequencing to analyze the biomarkers of the equine osteoarthrosis and to verify the efficacy of miR-92a in vitro.
I have specified a few questions below:
1. About animal breeds, there were four breeds in the mild OA group and nine in the severe OA group, without any overlapping strains. What is the reasonability of this choice?
2. In Figure 1, two groups of mild and severe OA could not be clustered together, and the two most different samples belonged to the same group (mild OA). In this case, the conclusions from the miRNA-sequencing will be quite limited.
3. In Figure 3, the title or the vertical coordinates of the graph can be written with the name of the miRNA to make it easier for the reader to distinguish.
Author Response
Thank you for taking the time to review our study.
1. About animal breeds, there were four breeds in the mild OA group and nine in the severe OA group, without any overlapping strains. What is the reasonability of this choice?
The study relied on collection of clinical samples donated by horse owners to an equine hospital over a set period of time. The population consisted of a wide range of horse breeds and there was no control over which breed was categorised into which group. It was not intentional to have no overlapping breeds but after classification the only breed that was included in both groups was the Thoroughbred. Future research involving a larger sample size with overlapping breeds would be beneficial. There have been other equine miRNA studies involving multiple breeds of horses which have shown the breeds of horses haven’t influenced miRNA expression (Castanheira et al 2021 and Lecchi et al 2018).
A comment about the influence of breeds has been added to the discussion, lines 383-388.
References:
- Castanheira, C., P. Balaskas, C. Falls, Y. Ashraf-Kharaz, P. Clegg, K. Burke, Y. Fang, P. Dyer, T. J. M. Welting and M. J. Peffers (2021). "Equine synovial fluid small non-coding RNA signatures in early osteoarthritis." BMC Vet Res 17(1): 26.
- Lecchi, C., E. Dalla Costa, D. Lebelt, V. Ferrante, E. Canali, F. Ceciliani, D. Stucke and M. Minero (2018). "Circulating miR-23b-3p, miR-145-5p and miR-200b-3p are potential biomarkers to monitor acute pain associated with laminitis in horses." Animal 12(2): 366-375.
2. In Figure 1, two groups of mild and severe OA could not be clustered together, and the two most different samples belonged to the same group (mild OA). In this case, the conclusions from the miRNA-sequencing will be quite limited.
We thank the reviewer for this comment and agree that Figure 1 illustrates that further analysis will reveal few differentially expressed molecules which was indeed the case. Additionally, unsupervised clustering and visualisation of the miRNA-seq data as a heatmap revealed no outlying samples and largely overlapping miRNA profiles between the mild and severe OA groups. However, three miRNAs were found to be significantly down regulated in the severe OA synovial fluid samples compared to the mild synovial fluid samples and so the RNA-seq data was useful to direct subsequent in vitro experiments.
3. In Figure 3, the title or the vertical coordinates of the graph can be written with the name of the miRNA to make it easier for the reader to distinguish.
Thank you for highlighting this, the figure has been amended.
Reviewer 3 Report
Osteoarthritis (OA) is the most prevalent health problem in ageing horses. The study verified that the miRNAs would demonstrate different differential expression profiles in mild OA synovial fluid samples compared to severe OA samples of the equine DIPJ. It’s a good topic but some contents still need to be revised before publication.
1. Please specify whether the breeds of horses affect the experimental results and prove it through more experiments if necessary.
2. Please specify how to distinguish mild OA and severe OA in horses.
3. There were not enough samples in the mild OA cohort.
Author Response
1. Please specify whether the breeds of horses affect the experimental results and prove it through more experiments if necessary.
Thank you for taking the time to review our study and for highlighting this interesting point. The results did not show that any particular breed influenced the results of miRNA expression in the RNA-seq data and the breeds used for the in vitro chondrocyte transfection experiments included two Irish Draught's, a cross breed pony, a Cob, a Cob cross and an Irish Sports Horse and there were no significant differences in gene expression between the donors. When the data from all donors was combined there was a significant increase in COMP, SOX9, RUNX2 and COL1A2 expression in the miR-92a mimic treatment compared to the negative control and similar expression of ACAN and COL3A1 in all three treatments across all donors.
A comment about the influence of breeds has been added to the discussion, lines 383-388, along with citing literature from other equine miRNA studies involving multiple breeds of horses which have shown the breeds of horses haven’t influenced miRNA expression (Castanheira et al 2021 and Lecchi et al 2018).
References:
- Castanheira, C., P. Balaskas, C. Falls, Y. Ashraf-Kharaz, P. Clegg, K. Burke, Y. Fang, P. Dyer, T. J. M. Welting and M. J. Peffers (2021). "Equine synovial fluid small non-coding RNA signatures in early osteoarthritis." BMC Vet Res 17(1): 26.
- Lecchi, C., E. Dalla Costa, D. Lebelt, V. Ferrante, E. Canali, F. Ceciliani, D. Stucke and M. Minero (2018). "Circulating miR-23b-3p, miR-145-5p and miR-200b-3p are potential biomarkers to monitor acute pain associated with laminitis in horses." Animal 12(2): 366-375.2.
2. Please specify how to distinguish mild OA and severe OA in horses.
The horses were categorised into either the mild OA or severe OA group based on macroscopic and microscopic scores. There are no reports of a specific articular cartilage macroscopic grading system for the equine DIPJ in the literature, therefore a system was developed and validated to grade the joints in the study. The scoring system included particular gross observations applicable to the DIPJ and adapted elements from McIlwraith et al 2010 scoring system of the equine metacarpo/metatarsophalangeal joint. There have not been any specific equine microscopic grading systems detailed in the literature and most assessments involve modified versions of the Mankin (Mankin et al. 1971) and OARSI systems (Pritzker et al. 2006). The OARSI system was developed for human cartilage assessment in 2006 and the system has been validated for animal (rats and sheep) (Custers et al. 2007) and human articular cartilage (Pearson et al. 2011; Pauli et al. 2012). It is now a recognised standard method of assessment for OA in human articular cartilage and therefore the OARSI system was chosen for use in the assessment of equine cartilage in this research.
The horse was classified as mild OA if the total gross joint grade of both the two feet per horse was <-6 and the combined OARSI grade of both feet was <-3. The horse was classified as severe OA if the total gross joint grade of both the two feet per horse was ->7 and the combined OARSI grade of both feet was ->3.
More detail about the macroscopic scoring system has been added to the experimental methods section in 3.2. Macroscopic and Microscopic Cartilage Assessment to clarify the classification for the reader and the mean scores have been removed. (Lines 417-431).
References
- Custers, R. J., L. B. Creemers, A. J. Verbout, M. H. van Rijen, W. J. Dhert, and D. B. Saris. 2007. 'Reliability, reproducibility and variability of the traditional Histologic/Histochemical Grading System vs the new OARSI Osteoarthritis Cartilage Histopathology Assessment System', Osteoarthritis Cartilage, 15: 1241-8.
- Mankin, H. J., H. Dorfman, L. Lippiello, and A. Zarins. 1971. 'Biochemical and metabolic abnormalities in articular cartilage from osteo-arthritic human hips. II. Correlation of morphology with biochemical and metabolic data', J Bone Joint Surg Am, 53: 523-37.
- McIlwraith, C. W., D. D. Frisbie, C. E. Kawcak, C. J. Fuller, M. Hurtig, and A. Cruz. 2010. 'The OARSI histopathology initiative - recommendations for histological assessments of osteoarthritis in the horse', Osteoarthritis Cartilage, 18 Suppl 3: S93-105.
- Pauli, C., R. Whiteside, F. L. Heras, D. Nesic, J. Koziol, S. P. Grogan, J. Matyas, K. P. Pritzker, D. D. D'Lima, and M. K. Lotz. 2012. 'Comparison of cartilage histopathology assessment systems on human knee joints at all stages of osteoarthritis development', Osteoarthritis Cartilage, 20: 476-85.
- Pearson, R. G., T. Kurien, K. S. Shu, and B. E. Scammell. 2011. 'Histopathology grading systems for characterisation of human knee osteoarthritis--reproducibility, variability, reliability, correlation, and validity', Osteoarthritis Cartilage, 19: 324-31.
- Pritzker, K. P. H., S. Gay, S. A. Jimenez, K. Ostergaard, J. P. Pelletier, P. A. Revell, D. Salter, and W. B. van den Berg. 2006. 'Osteoarthritis cartilage histopathology: grading and staging', Osteoarthritis and Cartilage, 14: 13-29.
3. There were not enough samples in the mild OA cohort.
The sample number for the mild OA cohort was smaller than the severe OA cohort which is a limitation encountered in clinical research. The study involved collection of samples from a mixed population of horses donated to an equine hospital in a set period and there was no control over which OA group their DIPJs would be classified to. A larger sample size and ideally the same number of samples in each group would be beneficial in future research. A power calculation was performed using our small RNA-seq data which showed a sample size of n=36 would be needed to demonstrate a power of 95% (significance level = 0.05) between the two groups for miR-92a.
A comment about the sample size has been added to the discussion, lines 278-284.
Round 2
Reviewer 2 Report
The revised version improves a lot.